# Cultivable and Non-Cultivable Approach to Bacteria from Undisturbed Soil with Plant Growth-Promoting Capacity

**DOI:** 10.3390/microorganisms13040909

**Published:** 2025-04-16

**Authors:** Lorena Jacqueline Gómez-Godínez, Pedro Cisneros-Saguilán, Dulce Darina Toscano-Santiago, Yair Eduardo Santiago-López, Saúl Neftalí Fonseca-Pérez, Magali Ruiz-Rivas, José Luis Aguirre-Noyola, Gabriel García

**Affiliations:** 1Centro Nacional de Recursos Genéticos, Instituto Nacional de Investigaciones Forestales, Agrícolas y Pecuarias, Boulevard de la Biodiversidad #400, Tepatitlán de Morelos 47600, Jalisco, Mexico; aguirre.jose@inifap.gob.mx; 2Programa de Maestría en Producción Agroalimentaria, Tecnológico Nacional de México Campus Instituto Tecnológico de Pinotepa, Santiago Pinotepa Nacional 71600, Oaxaca, Mexico; m23730314@pinotepa.tecnm.mx; 3Programa de Ingeniería en Agronomía, Tecnológico Nacional de México Campus Instituto Tecnológico de Pinotepa, Santiago Pinotepa Nacional 71600, Oaxaca, Mexico; l20730042@pinotepa.tecnm.mx (D.D.T.-S.); l20730257@pinotepa.tecnm.mx (Y.E.S.-L.); 4Instituto Nacional de Investigaciones Forestales, Agrícolas y Pecuarias, Campo Experimental Uruapan, Uruapan 60150, Michoacán, Mexico; ruiz.magali@inifap.gob.mx; 5Centro de Bachillerato Tecnológico Agropecuario No. 10, Santiago Pinotepa Nacional 71600, Oaxaca, Mexico; ggarciaiias@gmail.com

**Keywords:** soil quality, microbial resources, mineralization, *zea mays* L., tropics

## Abstract

Undisturbed soils are essential ecosystems with high microbial diversity. Microorganisms present in the soil can regulate biogeochemical cycles, making available and transforming different minerals in the soil, such as nitrogen, phosphorus and sulfur. In this study, the microbiota of undisturbed soils was characterized using an integrated approach of 16S rRNA ribosomal gene amplicon analysis and classical microbiology techniques. Phylum-level analyses revealed a high abundance of Proteobacteria, Acidobacteria, Verrucomicrobia and Actinobacteria, key groups in nutrient recycling, organic matter decomposition and plant-microorganism interaction. In the genus analysis, *Nitrospira* spp., *Candidatus Koribacter* spp., *Burkholderia* spp., *Bacillus* spp., *Flavobacterium* spp. and *Pedomicrobium* spp. were identified, with important functions in nitrification, plant growth promotion, organic matter degradation, and recovery of degraded soils. On the other hand, by using selective and differential media, it was possible to demonstrate the presence of microorganisms such as *Enterobacter* spp. and *Hafnia* spp., with the ability to solubilize phosphorus and potassium and produce siderophores, which are likely contributing to the biogeochemical cycles and plant growth within the soil studied.

## 1. Introduction

Soils are complex ecosystems that host an enormous microbial diversity, which has fundamental functions in maintaining ecosystem services such as food production, climate regulation and pest control [1]. These microbial communities regulate biogeochemical cycles and participate in the decomposition of organic matter and the availability of essential nutrients for plants, which are critical factors for the sustainability of terrestrial ecosystems [2,3].

Soil microbiota includes bacteria, archaea and fungi; of these microorganisms, bacteria are the most common and represent approximately 95% of the microbiota; around 10^8^ to 10^9^ cells exist in one gram of soil. The amount of cultivable bacterial cells in the soil is around 1% to 3% of the total cells [4,5]. These microorganisms interact with each other and the environment [6,7]. These interactions contribute to key processes such as nitrogen fixation, phosphate solubilization and phytohormone production [8,9]. Within this microbiota, plant growth-promoting bacteria (PGPB) play a prominent role by enhancing plant development through direct and indirect mechanisms [5,10,11]. Among the genera potentially promoting plant growth are *Bacillus*, *Burkholderia*, *Enterobacter*, *Hafnia*, *Nitrospira* and *Flavobacterium*, which can make nutrients available to the plant under normal or stress conditions. They are also of great interest to both ecology and biotechnology due to their potential to increase agricultural productivity sustainably [12,13,14,15,16,17].

Microorganism diversity and functionality depend on several factors; for example, soil type is one of the main factors shaping the root microbiome [18,19,20]. Microorganisms are sensitive to soil disturbances and management practices, which can lead to changes in their functionality [21]. For example, it has been reported that the relative abundance of Ascomycota increases when wheat and rice crop soil is treated with organic fertilizer. In contrast, the relative abundance of Zygomycota increases when the soil is treated with chemical fertilization [22]; it has also been reported that the relative abundance of Acidobacteria is significantly higher in natural forests compared to agricultural soils [23]. Land use changes such as the conversion of natural vegetation to conventional agriculture generate alterations in plant cover, litter and root biomass, as well as a decrease in soil organic carbon, which can significantly affect soil physico-chemical properties and drive changes in the composition and diversity of microbial communities, leading to soil degradation and loss of diversity [24,25,26,27]. In particular, undisturbed soils offer unique conditions to study microbial communities in their natural state, as they are free from disturbances caused by human activities such as intensive agriculture or urbanization [28].

The development of DNA sequencing technologies has revolutionized the study of soil microbiota. This has allowed the study of non-culturable microorganisms. Amplicon analysis, also called metabarcoding, is based on the amplification of variable regions of DNA or marker genes, such as 16S rRNA for bacteria. Amplicon sequencing allows us to study and describe cultivable and non-cultivable microorganisms within soil microbial communities. This technology represents the most cost-effective and effective sequencing method for studying these communities [29,30,31,32]. However, complementing these studies with classical microbiology techniques, such as isolation, cultivation of microorganisms and identification of biotechnological and agricultural potential, such as solubilization of phosphorus and potassium and production of siderophores, is essential to validate the results and explore practical applications of the identified microorganisms [8].

The objective of this study was to describe the microbiota associated with undis-turbed soil using classical microbiological techniques such as isolation and morpho-logical and biochemical characterization, and the identification of plant growth-promoting microorganisms. Furthermore, the microbiota was also described using a non-culturable metabarcoding approach. This integrated approach not only deepens the knowledge of microbial diversity but also identifies key taxa with ecological functions and biotechnological applications. The results contribute to understanding the processes that maintain the stability of terrestrial ecosystems and highlight the importance of conserving undisturbed soils as reservoirs of biodiversity and biological resources.

## 2. Materials and Methods

### 2.1. Description of the Study Area and Soil Sample

The experimental site for this study is located in San José de las Flores, municipality of Santiago Jamiltepec, Oaxaca, Mexico (16°24′38.4″ N and 97°44′20.5″ W, 625 masl). It has a warm subhumid climate with summer rains (with temperature and precipitation ranges between 22 and 26 °C and 1500 and 2000 mm), mountainous relief with slopes > 25%, shallow intrusive rock soils and forest-type vegetation (Figure 1); these climatic, edaphic and orographic conditions allow the production of crops such as coffee, corn, beans, squash, hibiscus and tomato [33]. However, nothing has been cultivated in the study area. Samples were collected in July 2024. Before collection, a sampling area of 900 m^2^ (18 m by 50 m long) was delimited (Figure 1), and the surface of each sampling area was manually cleaned to remove organic material such as leaf litter, branches, plant stems and inorganic debris. Three zigzag transects were made on the farm (each transect on the farm corresponds to a sample/replicate, designated as sample 1, sample 2 and sample 3) (Figure 1). A sample of 50–150 g of soil was collected, per plotted point (nine points per sample), using a sterile metal spatula at 0–30 cm depth. Samples were stored in sterile Ziploc-type plastic bags (4 °C) and transported to the Laboratory for further processing.

### 2.2. Isolation of Culturable Bacteria

Serial dilutions were made to obtain the bacterial isolates from the soil sampled. 10 g of soil was weighed and poured into a flask containing 100 mL of saline solution. It was shaken, and four test tubes containing 10 mL of saline solution were placed on a rack. Dilutions were made up to 10^−5^ (NOM, 110-SSA1.1994) [34]. A total of 0.1 mL from the dilutions 10^−3^, 10^−4^ and 10^−5^ was plated on sterilized tryptic agar plates and prepared according to the manufacturer’s instructions. The media were incubated for 24 h at 28 ± 2 °C. The isolated and pure cultures were maintained in nutrient agar to subsequently proceed to their morphological and biochemical characterization.

### 2.3. Morphological and Biochemical Characterization of Rhizobacteria

A morphological characterization was performed on the isolated, pure and viable microorganisms. Characteristics such as shape, size and pigmentation were considered to describe the morphology of the bacterial colonies. Gram staining was used to evaluate bacterial morphology, allowing us to identify Gram-positive and Gram-negative bacteria. To determine part of the metabolism and achieve a biochemical characterization, tests such as the catalase and oxidase were performed; MIO medium was used to evaluate Motility, Indole, and Ornithine; LIA medium was used to identify the capacity to deaminate or decarboxylate lysine and the production of hydrogen sulfide; OF test is a biochemical test that determines the type of energy metabolism of bacteria; and the MR/VP test is used to identify bacteria and determine the glucose fermentation pathway; these tests were carried out using the standardized procedure as described by Clarke and Cowan [35].

### 2.4. Phosphate Solubilization

A spot inoculation of each bacteria was carried out with phosphate solubilization screening isolate on Pikovskaya’s media [36]. The Pikovskaya agar was composed of 5 g of tricalcium phosphate (Ca_3_(PO_4_)_2_), 10 g of glucose (C_6_H_12_O_6_), 0.002 g of manganese sulfate (MnSO_4_·H_2_O), 0.2 g of sodium chloride (NaCl), 0.2 g of potassium chloride (KCl), 0.1 g of magnesium sulfate (MgSO_4_), 0.5 g of ammonium sulfate ((NH_4_)_2_SO_4_), 0.5 g of a yeast extract, 15 g of agar, and 1000 mL of sterile distilled water at pH 7.0. The media were sterilized, and the microorganisms were inoculated when they reached room temperature. The microorganisms were spot inoculated into the center of the Petri dishes, which were placed in an incubator for 4 to 5 days at a temperature of 28 ± 2 °C. After incubation, the dishes were read, indicating a positive result if a clear zone appeared around the bacterial colony.

### 2.5. Potassium Solubilization

A spot inoculation of each bacteria carried out potassium solubilization screening isolate on modified Pikovskaya’s media [37].

### 2.6. Siderophore Production

To determine siderophore production, chromium-azurol S (CAS) medium was used, consisting of 10 mL of a Fe(III) solution (27 mg of FeCl_3_ 6H_2_O and 83.3 µL of concentrated HCl in 100 mL of ddH_2_O) together with 72.9 mg of hexadecyltrimethylammonium bromide (HDTMA). Each microorganism was incubated for 24 h, and then, with a sterile loop, it was seeded in the CAS medium. The plates were incubated at 37 °C for between 48 and 72 h. The appearance of a yellow/orange color around the colonies was identified as a positive result for siderophore production [38].

### 2.7. Jensen’s N-Free Medium

To determine whether bacteria could grow under nitrogen-deficient conditions, Jensen’s medium was used, consisting of 20 g; K_2_HPO_4_ at 1.0 g; MgSO_4_·7H_2_O at 0.5 g; NaCl at 0.5 g; FeSO_4_·7H_2_O at 0.1 g; CaCO_3_ at 2.0 g; agar at 15.0 g; Na_2_MoO_4_ at 0.005 g; and 1.0 L of sterile distilled water [39]. The medium was adjusted to a pH of 7.2 and then sterilized (121 °C for 15 min). The microorganisms to be evaluated were inoculated into Petri dishes with Jensen’s medium and incubated for 7 days at a temperature of 28 ± 2 °C. At the end of the incubation period, the growth of the microorganisms was verified, and those that could grow under these conditions were considered positive.

### 2.8. Growth Promotion in Corn Seedlings by Promoter Bacteria

Corn seeds of the H-391 hybrid were externally disinfected according to the method of Gómez-Godínez et al. [40]. The ISTA germination method was modified and used to evaluate the microorganisms’ growth-promoting capacity [41]. Ten seeds were placed in an alternating row between sterile cellulose paper, at the bottom of which 10 mL of a bacterial suspension at 1 × 10^8^ CFU mL^−1^ was applied. Each paper containing the seeds was rolled up and incubated in a growth chamber under a photoperiod of 14 h of light/10 h of darkness, with 75% humidity and 25 °C. Each treatment was replicated three times; seeds with 10 mL of distilled water were used as a control. Fifteen days post-inoculation, various parameters were measured, such as root length, shoot length, fresh and dry weight of shoots and fresh and dry weight of roots. Data were evaluated using analysis of variance (ANOVA) and Tukey’s mean separation test (*p* ≤ 0.05).

### 2.9. Description of No Culture Soil Bacteria

#### 2.9.1. DNA Extraction and Preparation of Libraries and Sequencing Procedure Metabarcoding Approach

From 0.2 g of each collected sample, metagenomic DNA was extracted using a commercial extraction kit (Fecal DNA Extraction Kit, Bio Basic Inc., Markham, ON, Canada). The integrity of the extracted DNA was verified using a 1% agarose gel, and it was stored at −20 °C until sequencing.

The construction of 16S DNA libraries started with the PCR-based amplification of hypervariable regions V2, V3, V4, V6–V9 of the 16S rDNA gene, achieved in two independent reactions throughout the use of the 16S metagenomics system following the manufacturer’s instructions (Thermo Fisher Scientific, Waltham, MA, USA) in a SelectCycler device (Select BioProduct, Life Science Research, Waltham, MA, USA). To generate 16S rDNA libraries using the commercial Ion Plus Fragment Library system and Ion Xpress barcoded adapters (Thermo Fisher Scientific), 50 nanograms of the equimolar mixture prepared from the amplification products were used. The Agentcourt AMPure XP system (following the manufacturer’s instructions: Beckman Coulter, Brea, CA) was used for library purification. Quantification was then performed using the Bioanalyzer 2100 (Agilent Technologies, Santa Clara, CA, USA) to achieve a 26 pM buffer. This was followed by emulsion PCR amplification using a 25 µL volume of the equimolar mixture of all samples (One-Touch 2, Thermo Fisher Scientific) and enriched with the OneTouch Enrichment system (Thermo Fisher Scientific). Sequencing was performed using the Ion S5™ system (Thermo Fisher Scientific).

#### 2.9.2. Bioinformatics Analysis

The sequencing files were converted to FASTQ. Quality assessment was performed using FastQC software (v0.12.0) [42]. Subsequent procedures, such as quality control and cleanup, were performed with QIIME2 (v.2019.7) [43]. Adapters and sequencing barcodes were removed using the Cutadapt tool (v. 2.6) [44]. Phred quality was used to evaluate sequences with a quality of Q30, leaving only those above this parameter. Low-quality sequences and chimeras were eliminated using the DADA2 tool (v.2019.7) [45]. Finally, clean sequences with a quality above 30 were compared to the Greengenes database (gg-13-8-99-515-806-nb-classifier.qza) to allow assignment to the Amplicon Variant Sequence (ASVs).

### 2.10. Evaluating Co-Occurrence Networks Between Genera

The Spearman correlation matrix was calculated from the relative abundance data (prior analysis of normality of the variable) to evaluate the co-occurrence relationships between microbial taxa. The correlation matrix was obtained using the correlation function in R package, and the Spearman correlation method and correlation threshold were applied to retain only those relationships with absolute values greater than 0.5 (Spearman’s ρ > |0.5|), according to what was suggested by Liu et al. [46]. The co-occurrence network and visualization were built using the Igraph library (v.1.2.6) in R (R version 3.6.1) [47]. The nodes with a high degree were considered as key species in microbial networks [48].

## 3. Results and Discussion

Twenty-nine bacteria were isolated from the undisturbed soil, which we will call US-B and isolated number (Undisturbed Soil Bacteria). According to the description method of Wood and Krieg [49], they had different colonial morphologies, including circular to irregular shapes, with entire and curly edges, and with and without shine (Figure 2).

### 3.1. Growth Promotion In Vitro Evaluation

From the evaluations in the different culture media, it was possible to evidence the solubilization of phosphorus (PS), potassium (KS), growth in nitrogen-limiting medium (NF) and production of siderophores (Sid); some isolates such as US-B15, presented all the growth promotion activities evaluated; on the other hand, isolates US-B10, US-B15 and US-B22 presented three growth promotion characteristics (PS, Sid and NF), isolates US-B21, US-B23 and US-B24 presented three growth promotion characteristics (Sid, NF and KS). Plants and microorganisms need iron for different cellular reactions, which they acquire from the medium they develop. Siderophores are organic molecules that form chelates with ferric ions and other metals such as molybdenum, manganese and zinc [50]. Different microorganisms can produce siderophores and make metals available to plants; an example of this is the Pseudomonas US-B10 strain, capable of producing siderophores, favoring the growth and yield of different crops such as barley and flax [51]. On the other hand, corn seeds inoculated with siderophore-producing Pseudomonas strains showed better iron absorption under iron stress conditions, and a significant increase in the germination percentage and plant growth was identified [52]. Finally, siderophores have been identified as inhibiting pathogens’ growth under in vitro, soil and greenhouse conditions [53,54].

Some microorganisms, such as *Enterobacter*, have been reported to be capable of solubilizing potassium through the production of organic acids [55,56]. One of the primary nutrients for plant growth is nitrogen; there are microorganisms responsible for transforming atmospheric nitrogen into assimilable nitrogen, such as Alphaproteobacteria (*Rhizobia*, *Bradyrhizobia*, *Rhodobacteria*), Betaproteobacteria (*Burkholderia*, *Nitrosospira*), Gammaproteobacteria (*Pseudomonas*, *Xanthomonus*), Firmicutes and Cyanobacteria [57,58,59] in the different ecosystems. Some isolates only presented one growth-promoting characteristic and isolates US-B26, US-B27, US-B28 and US-B29 did not present a single characteristic (Table 1).

### 3.2. Identification of PGPB from Biochemical Tests

Bacteria that presented three or more growth-promoting characteristics were identified through their biochemical characteristics. Oxidase-negative bacteria were found, indicating they do not have the oxidase enzyme. Most isolates were catalase-positive, which shows the presence of an enzyme that is capable of breaking down hydrogen peroxide into water and oxygen. Some could metabolize carbohydrates through oxidative and fermentative pathways. Most isolates had the ability to ferment sugars other than glucose. However, most isolates could not ferment lactose. Except for isolate US-B24, all isolates showed mobility. Isolates US-B10 and US-B15 were identified as *Enterobacter* spp., while isolates US-B22 and US-B24 belong to the genus *Providencia* spp.; according to biochemical tests, isolate US-B21 is *Hafnia* spp., and isolate US-B23 is *Aeromonas* spp.

### 3.3. Evaluation of Growth Promotion in Corn Seedlings

The strains analyzed did not show a significant difference in root growth (*p* > 0.05). However, the growth of the aerial part was favored by strain US-B15 by 8%, US-B21 by 30% and US-B22 by 27% more compared to non-inoculated plants (*p* < 0.05) (Figure 3). These isolates were identified through their biochemical tests. US-B15 was identified as *Enterobacter* spp.; different species of this genus are known to have different plant growth-promoting capacities, such as nitrogen fixation, phosphate solubilization, Indole Acetic Acid production and tolerance to abiotic stress [60,61,62]. *Enterobacter* sp. DBA51 has been identified with the ability to promote height in tomato (*Solanum lycopersicum* L.) and tobacco (*Nicotiana tabacum* L.) plants by up to 20% and 40% gain in root biomass, compared to non-inoculated plants [63]. On the other hand, *Enterobacter* sp. J49, inoculated individually or with chemical fertilizers, favors the growth of corn plants under field conditions [64]. The isolate US-B21 was identified as *Hafnia* spp.; species belonging to this genus have been isolated from the rhizosphere, and its genome has presented genes related to the production of siderophores and promotion of plant growth [65]. Finally, isolate US-B22, identified as Providence spp., favored 27% of the plant’s leaf growth. These bacteria can generate defense enzymes in wheat plants (*Triticum aestivum*) in field conditions, as well as the capacity to produce Indole Acetic Acid, ammonium and the solubilization of phosphate and zinc [66].

### 3.4. Analysis of the Non-Cultivable Microbiota Associated with Undisturbed Soil

Quality sequences were obtained from sequencing samples. Sample 1 had 48,815 reads, sample 2 had 54,130 and sample 3 had 50,147 reads. Amplicon analysis revealed a high microbial diversity in the undisturbed soil, with a predominance of the phyla Proteobacteria, Acidobacteria, Verrucomicrobia and Actinobacteria in average percentages of 56%, 17%, 9% and 8%, respectively (Figure 4). The sequences were deposited in NCBI under the number SUB15219394. Hua et al. [23] reported a similar predominance of Proteobacteria (27%), Acidobacteria (18%), Actinobacteria (16%), and Chloroflexi (8%) specifically for a native forest in China. In another study, for an undisturbed soil in Thailand sampled over a year through the summer, rainy and winter seasons, Arunrat et al. [21] reported the following predominance: Proteobacteria (26%), Actinobacteria (23%), Planctomycetes (20%), Acidobacteria (13%) and Verrucomicrobia (9%). Meng et al. [25], in a comparative study among forest types (Native, Bamboo, Fir and Mixed) in China, reported a predominance of Acidobacteria (49%), Proteobacteria (33%), Actinobacteria (6%) and Verrucomicrobia (3%) on average for the native forest.

Proteobacteria are one of the most abundant groups in nature, consisting of 460 genera and more than 1600 species, covering many Gram-negative bacteria and being important in medical, industrial, veterinary and agricultural areas. Specifically in the agricultural area, Proteobacteria are associated with key processes such as nitrogen fixation, decomposition of organic matter and plant–microorganism interaction, which makes them essential actors in nutrient cycling and soil functioning [67,68]. Acidobacteria are widely distributed in diverse natural environments and play important roles in various soil ecological processes, such as the decomposition of organic compounds and nutrient cycling [69,70]. Verrucomicrobia are distributed in different environments such as water bodies, landfill leachates, animal and human intestines and soil from 16S rRNA sequences; they have been identified in amounts from 0 to 21%. However, this wide diversity is poorly represented in pure cultures, finding only 12 described genera, most of them being aerobic, neutrophilic and chemoorganoheterotrophic [71,72,73].

At the genus level, a high percentage of *Pigmentiphaga* 34%, *Rhodoplanes* 26%, *Candidatus Solibacter* 10%, *Kaistobacter* 6%, *Burkholderia* 3% and *Candidatus Koribacter* 3% and some other genera with lower percentages, such as *Gluconacetobacter*, *Ktedonobacter*, DA101, *Dyella* and *Bacillus* were identified (Figure 5). In their case, Hua et al. [23] reported a predominance of genera (relative abundance greater than 5%), consisting of *Subgroup 2_norank* (11.69%), *Gaiellales_norank* (6.6%), *Gemmatimonadaceae_uncultured* (6.21%), *Acidobacteriales_norank* (5.94%) and *Bradyrhizobium* (5.13%), in the soil of a native forest in China. Arunrat et al. [21] reported a low but stable distribution (0.5 to 7% relative abundance) of *Candidatus Udaeobacter*, *Bacillus*, *Conexibacter*, *Bradyrhizobium*, *Candidatus Xiphinematobacter*, *Acidothermus*, *Geodermatophilus*, *HSB OF53-F07 and Gemmata*, during a year at several sampling times (summer, rainy and winter) for an undisturbed soil in Thailand.

Blümel et al. [74] described for the first time the genus *Pigmentiphaga*, which is a Gram-negative, facultatively anaerobic, oxidase and catalase-positive bacterium; it has been isolated from soils amended with humic acids [75]. It has been reported that this genus can degrade neonicotinoid insecticides in contaminated soils [76]; furthermore, it has been identified that these bacteria can degrade some autotoxic allelochemicals and xenobiotics and, thus, favor the percentage of germination and development of tobacco stems, roots and leaves [77]. Bacteria of the genus *Rhodoplanes* have been isolated from the activated sludge and are characterized by phototrophic bacteria that can carry out complete denitrification [78]. In corn conservation tillage soils, the abundance of these bacteria has increased, and their function is involved in nitrate reduction and denitrification processes [79]; this is probably the function in undisturbed soil. From soil samples, it was possible to compare different Acidobacteria genomes, identifying some of them as *Candidatus Solibacter* and *Candidatus Koribacter*; these organisms presented genes related to nitrate transport and genes related to the ability to produce siderophores [80,81].

The bacteria belonging to the Gluconacetobacter genus belong to the Acetobacteri-aceae family, which are aerobic and Gram-negative bacteria [82] that can be distributed in different environments such as grapes, flowers, fruits and bee hives, alcoholic beverages, fermented foods and soil [82,83,84]. In the soil, they develop functions such as biological nitrogen fixation [85,86]. Bacteria of the genus Ktedonobacter belong to the phylum Chloroflexi and have been identified in the rhizosphere of plants [54]. These bacteria can produce enzymes that can degrade carbohydrates, such as cellulose [87]. Therefore, their function in the soil could be the degradation of plant matter. The genus DA101 was described in 1998, representing a high abundance of its sequence in grassland soils, being found up to six times more than in forest soils [88,89]. Dyella is a genus of Gram-negative, rod-shaped, motile, non-spore-forming bacteria; this genus has been isolated from other types of undisturbed soil and favors the growth of tomato and Arabidopsis seedlings; its presence has also favored the growth of cucumber [90,91,92]. Bacteria of the genus Burkholderia live mainly in the soil and perform diverse ecological functions as saprophytes and nitrogen fixers [93]; it has also been reported that these bacteria can inhibit the symptoms of crown rot caused by Fusarium graminearum in wheat crops [94]. Finally, the Bacillus genus is one of the soil’s most predominant plant growth promoters [95]. These are Gram-positive bacteria; they can form endospores, which allows them to tolerate adverse conditions such as UV radiation and, thus, remain for long periods in the soil and act as biocontrollers [96,97]. This microorganism is cultivable, has a very broad biotechnological potential, and is present in this type of undisturbed soil. In general, these results suggest that the bacterial community of the soil analyzed is composed of genera with essential functions in ecosystem stability, nutrient recycling and the promotion of plant growth. The undisturbed nature of the soil has allowed the coexistence of these microorganisms, which reinforces their potential in agricultural applications as biofertilizers and biocontrol agents.

The variation in bacterial abundance and diversity at both the phyla and genus levels between this and other studies is explained by some environmental and soil management factors, as well as in some cases by the genotype of the cultivated species and the type of vegetation. Meng et al. [25] reported that some physicochemical properties of the soil, land use history and vegetation type affect the composition and diversity of bacterial communities. Peiffer et al. [19] revealed that corn genotype and field climatic conditions also affected the relative abundance and diversity of soil microbiota. Additionally, Arunrat et al. [21] pointed out that climatic conditions (summer, rainy and winter), physicochemical properties of the soil, soil management practices and vegetation cover affect soil bacterial composition and diversity.

The co-occurrence analysis reveals a highly interconnected structure with several key genera, among which *Pigmentiphaga*, *Candidatus Solibacter* and *Rhodoplanes* stand out (Figure 6). These genera present multiple connections with other genera, suggesting their central role in the dynamics of the microbial community and indicating that they could be playing a key ecological role in the stability of the microbial ecosystem. *Cupriavidus* and *Pedomicrobium* present fewer associations, suggesting they may be limited to specific environmental conditions or have more specialized interactions.

Arunrat et al. [21] demonstrated that undisturbed soils exhibit more stable bacterial diversity across seasons (summer, rainy and winter); this is due to their nutrient-rich soil, resulting from faster organic matter turnover. Also, due to the high soil-moisture content, undisturbed soils could maintain a balanced daily and seasonal soil temperature, creating a local microclimate favorable for soil bacteria.

## 4. Conclusions

The results obtained in this study confirm that undisturbed soils host a diverse and functionally relevant microbial community that maintains biogeochemical cycles and ecological sustainability. The high abundance of Proteobacteria, Acidobacteria, Verrucomicrobia and Actinobacteria highlights their key role in nutrient recycling and organic matter decomposition, essential processes for the stability of these ecosystems.

At the taxonomic level, the identification of families such as Alcaligenaceae, Hy-phomicrobiaceae, Koribacteraceae, Solibacteraceae and Sinobacteraceae suggests an in-tense activity of nitrogen fixation, degradation of organic compounds and bioremediation, which reinforces the importance of these soils as reservoirs of microorganisms with critical ecosystem functions. In addition, the presence of genera such as *Nitrospira*, *Candidatus Koribacter*, *Burkholderia*, *Bacillus*, *Flavobacterium* and *Pedomicrobium* highlights their role in key processes such as nitrification, promotion of plant growth and restoration of degraded soils.

The isolation of *Bacillus* as a plant growth-promoting bacterium represents an op-portunity for its application in sustainable agricultural practices, favoring productivity without compromising ecosystem integrity. These findings reinforce the need to conserve undisturbed soils, not only for their high biodiversity but also for their potential in bi-otechnological and ecological applications.

However, the study has some limitations, such as the lack of a more in-depth assessment of microbial interactions as a whole and the impact of long-term climatic factors on the microbial community. Future lines of research could focus on a more detailed characterization of the relationships between microorganisms and their functions in the context of different soil types, as well as on the impact of human activity and climate change on the stability of these ecosystems.

This study provides evidence of the value of undisturbed soils as strategic reservoirs of microbial biodiversity, underlining the importance of their protection and proper management for maintaining essential ecosystem services in environmental balance.

## Figures and Tables

**Figure 1 microorganisms-13-00909-f001:**
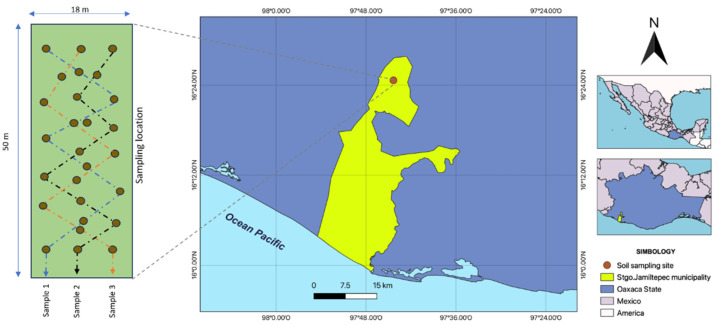
Location of the study site in San José de las Flores, Santiago Jamiltepec, Oaxaca.

**Figure 2 microorganisms-13-00909-f002:**
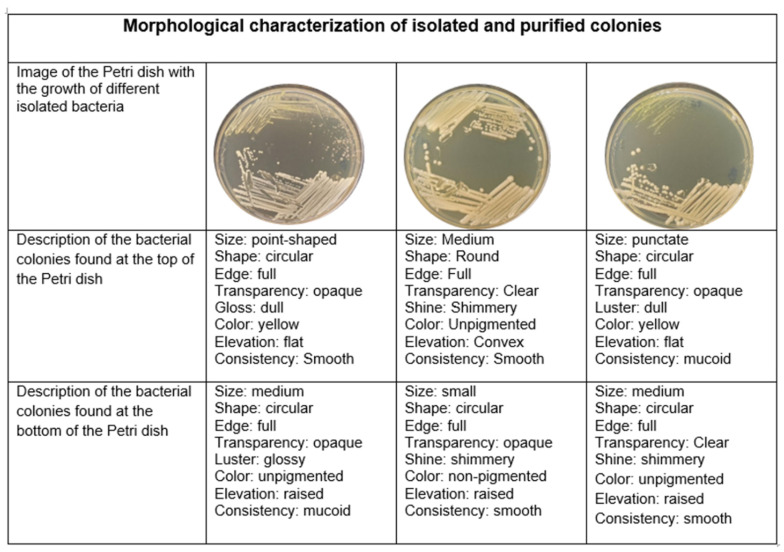
Description of the colony morphology of the different isolates in the undisturbed soil.

**Figure 3 microorganisms-13-00909-f003:**
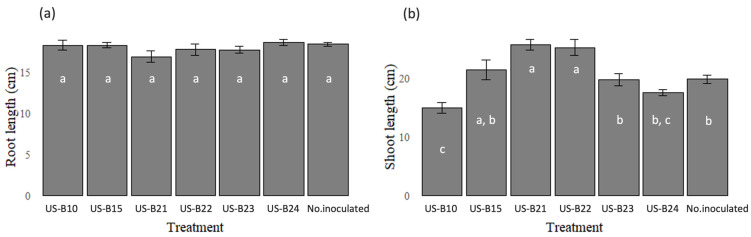
Evaluation of the effect of plant growth-promoting bacteria on corn seedlings. (**a**) Evaluation of root growth. (**b**) Evaluation of growth of aerial parts of corn plants. Values are mean of 10 replicates ± standard deviation of means, and the means with different letters between the bars indicate differences between treatments (*p* < 0.05).

**Figure 4 microorganisms-13-00909-f004:**
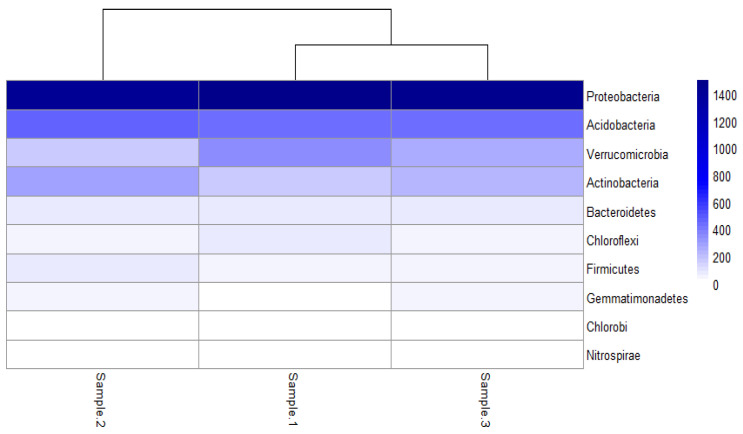
Relative abundances at the phylum level.

**Figure 5 microorganisms-13-00909-f005:**
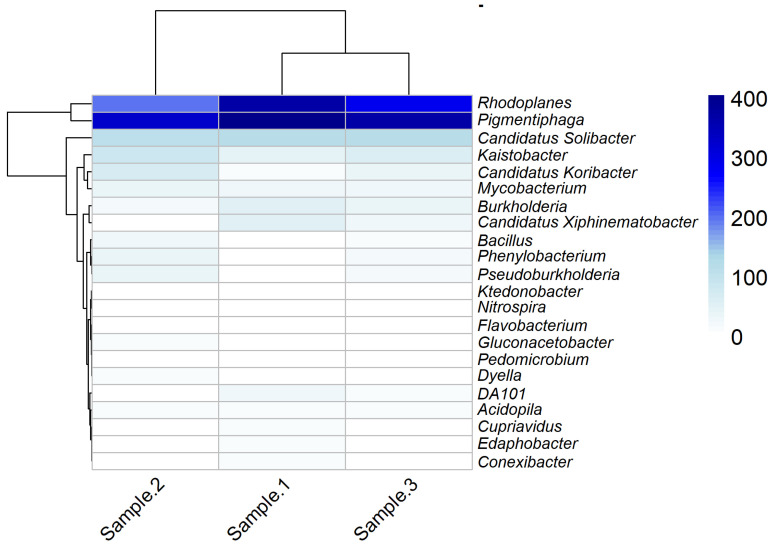
Relative abundance of the main genera detected in the soil samples.

**Figure 6 microorganisms-13-00909-f006:**
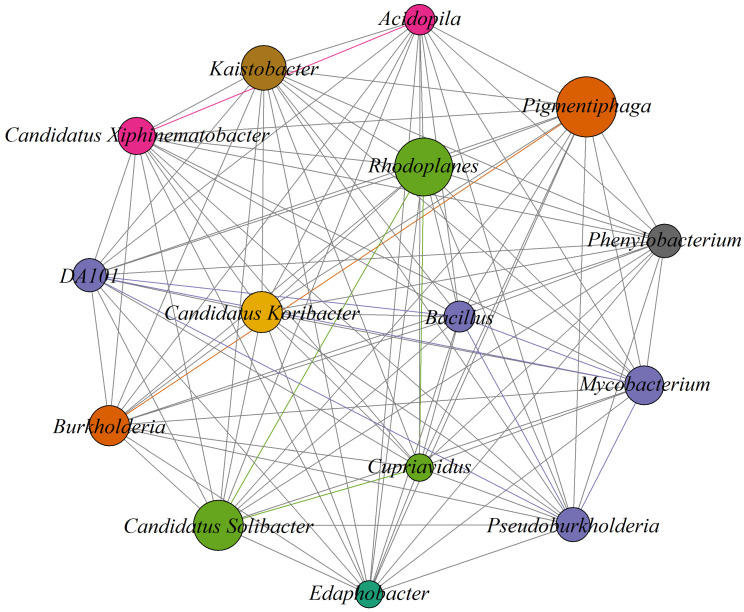
Relative abundance of the main genera detected in the soil samples. Each circle represents a different bacterial species (genus and species are indicated above each circle). Lines connect the circles, indicating a correlation between species (a threshold of 0.5 to 0.9 was used to select significant correlations).

**Table 1 microorganisms-13-00909-t001:** Functional characterization of bacteria.

Bacteria	Phosphorus Solubilization	Production of Siderophores	Jensen’s N-Free Medium	Potassium Solubilization
US-B1	−	−	+	+
US-B2	−	−	−	+
US-B3	−	−	−	+
US-B4	−	−	+	+
US-B5	−	−	+	+
US-B6	−	−	+	+
US-B7	−	−	+	+
US-B8	−	+	+	+
US-B9	−	−	+	−
US-B10	+	+	+	−
US-B11	−	−	+	−
US-B12	−	−	+	+
US-B13	−	−	+	+
US-B14	−	−	+	−
US-B15	+	+	+	+
US-B16	−	+	+	+
US-B17	−	−	+	−
US-B18	−	−	+	+
US-B19	−	−	−	+
US-B20	+	+	−	−
US-B21	−	+	+	+
US-B22	+	+	+	−
US-B23	−	+	+	+
US-B24	−	+	+	+
US-B25	−	+	−	−
US-B26	−	−	−	−
US-B27	−	−	−	−
US-B28	−	−	−	−
US-B29	−	−	−	−

Notes: + = Positive, − = Negative. Bacteria with more than two growth-promoting characteristics for the following tests were marked in gray.

## Data Availability

The original contributions presented in this study are included in the article. Further inquiries can be directed to the corresponding authors.

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
