# Peer review of "Cultivable and Non-Cultivable Approach to Bacteria from Undisturbed Soil with Plant Growth-Promoting Capacity"

_microorganisms, 2025, doi:10.3390/microorganisms13040909_

Round 1
Reviewer 1 Report
Comments and Suggestions for Authors
The paper has an interesting subject that remains in agreement with Microorganisms scope. This study highlights the importance of conserving undisturbed soils as reservoirs of biodiversity and as a source of microorganisms with ecological and biotechnological applications.
However evaluated paper demands some modification before potential acceptation.
My suggestions are as follows:
Line 47 – should be 108 and 109 instead of 108 and 109
The description of the study area is very vague. It is not explained what is meant by undisturbed soil area?
The authors, by writing that there are suitable conditions for cultivation i.e. coffee, corn, beans, pumpkin, hibiscus and tomato in the area imply, as it were, that such cultivation is being carried out there, so how can one speak of an undisturbed site?
There is lack of information about this site history which would reassure the reader that the area is undisturbed and suitable for the research the authors have described.
What size area was chosen for the study?
Lines 100-101 – sentence demands to be rewritten as the current form is stylistically incorrect and contains a repetition of the word obtained three times
Line 103 – should be 10-5
Line 104 – the same, be careful in using superscript
Line 163 – for what reason you decide to use fecal kit for DNA extraction form soil samples?
Line 217-219 – names of classes or phyla of bacteria should be written in capital letters, please correct this
Line 219 – names of genera should be written with italic style
Line 222 - isolate numbers tell the reader nothing, they need to be identified and assigned an appropriate taxonomy
Line 252 – why Zinc is written with capital letter?
Line 257 – what does it means sample 1…sample 2 etc.? Perhaps it would be useful to include a diagram with sampling points in the methodology? and include information on how many of these points have been selected?
Line 282 – phylum not Phylum
Line 304 – quality of Figure 5 is poor, the legend is completely unreadable
What is more, the previous text of the paper does not refer to these three samples at all - where did they come from, how are they different, why were these three chosen?
The lack of a precise history of the area surveyed does not allow the conclusion that it is intact...since when? and this calls into question the conclusions drawn. Without the addition of an adequate description, the paper is not suitable for publication. There is a lack of detailed description of the samples taken with a map of sampling etc. - was it just the 3 points only indicated in Figure 5? and each sample of 50-150 g allowed to infer biodiversity?
The paper has an interesting subject that remains in agreement with Microorganisms scope. This study highlights the importance of conserving undisturbed soils as reservoirs of biodiversity and as a source of microorganisms with ecological and biotechnological applications.
However evaluated paper demands some modification before potential acceptation.
My suggestions are as follows:
Line 47 – should be 108 and 109 instead of 108 and 109
The description of the study area is very vague. It is not explained what is meant by undisturbed soil area?
The authors, by writing that there are suitable conditions for cultivation i.e. coffee, corn, beans, pumpkin, hibiscus and tomato in the area imply, as it were, that such cultivation is being carried out there, so how can one speak of an undisturbed site?
There is lack of information about this site history which would reassure the reader that the area is undisturbed and suitable for the research the authors have described.
What size area was chosen for the study?
Lines 100-101 – sentence demands to be rewritten as the current form is stylistically incorrect and contains a repetition of the word obtained three times
Line 103 – should be 10-5
Line 104 – the same, be careful in using superscript
Line 163 – for what reason you decide to use fecal kit for DNA extraction form soil samples?
Line 217-219 – names of classes or phyla of bacteria should be written in capital letters, please correct this
Line 219 – names of genera should be written with italic style
Line 222 - isolate numbers tell the reader nothing, they need to be identified and assigned an appropriate taxonomy
Line 252 – why Zinc is written with capital letter?
Line 257 – what does it means sample 1…sample 2 etc.? Perhaps it would be useful to include a diagram with sampling points in the methodology? and include information on how many of these points have been selected?
Line 282 – phylum not Phylum
Line 304 – quality of Figure 5 is poor, the legend is completely unreadable
What is more, the previous text of the paper does not refer to these three samples at all - where did they come from, how are they different, why were these three chosen?
The lack of a precise history of the area surveyed does not allow the conclusion that it is intact...since when? and this calls into question the conclusions drawn. Without the addition of an adequate description, the paper is not suitable for publication. There is a lack of detailed description of the samples taken with a map of sampling etc. - was it just the 3 points only indicated in Figure 5? and each sample of 50-150 g allowed to infer biodiversity?
Comments on the Quality of English Language
English quality should be improved, there are many repetition of the same words in the text - even in one sentence
Author Response
The response to the comments from Reviewer 1 is addressed in the attached document.

Reviewer 2 Report
Comments and Suggestions for Authors
Minor comments:
(Lines 21–25): How does the microbiota of undisturbed soil contribute to nutrient cycling and ecological stability?
(Lines 29–31, 52–55): What are the main genera of plant growth-promoting bacteria (PGPB) identified in the study, and what roles do they play?
(Lines 57–63): How do land-use changes and soil disturbances affect the diversity and functionality of microbial communities?
(Lines 69–75, 161–172): What methodologies were used in the study to analyze both culturable and non-culturable microorganisms?
(Lines 130–137, 211–213): What is the significance of siderophore production in microbial interactions and plant growth promotion?
(Lines 206–224, 225–234): Which bacterial isolates demonstrated multiple plant growth-promoting activities, and how were they identified?
(Lines 149–159, 235–252): What was the impact of specific bacterial strains on the growth of corn seedlings, and how was it assessed?
(Lines 256–273, 276–299): What key bacterial taxa were identified in the metagenomic analysis, and what ecological roles do they play in undisturbed soil?
Author Response
The response to the comments from Reviewer 2 is addressed in the attached document.

Round 2
Reviewer 1 Report
Comments and Suggestions for Authors
Thank you for correction of the ms according my suggestions. Also thank you for answering on my comments, I accept your clarification.
Line 292 - correct proteobacteria into Proteobacteria
Comments on the Quality of English Language
Enqlish should be corrected before publication.
Author Response
The comments on the observations are in the attached document. Thank you very much.
